

# Uncovering hidden genetic risk factors for breast and ovarian cancers in BRCA-negative women: a machine learning approach in the Saudi population

Nofe Alganmi[1,2,3], Arwa Bashanfar[4], Reem Alotaibi[4], Haneen Banjar[1,2,3], Sajjad Karim[2,5], Zeenat Mirza[5,6], Heba Abusamra[2], Manal Al-Attas[2], Shereen Turkistany[7] and Adel Abuzenadah[2,5]

[1] Computer Science, Faculty of Computing and Information Technology, King Abdulaziz University, Jeddah, Saudi Arabia
[2] Center of Excellence in Genomic Medicine Research, King Abdulaziz University, Jeddah, Saudi Arabia
[3] Centre of Artificial Intelligence in Precision Medicines, King Abdulaziz University, Jeddah, Saudi Arabia
[4] Information Technology, Faculty of Computing and Information Technology, King Abdulaziz University, Jeddah, Saudi Arabia
[5] Department of Medical Lab Technology, Faculty of Applied Medical Sciences, King Abdulaziz University, Jeddah, Saudi Arabia
[6] King Fahd Medical Research Center, Faculty of Applied Medical Sciences, King Abdulaziz University, Jeddah, Saudi Arabia
[7] Center of Innovation Personalized Medicine, King Abdulaziz University, Jeddah, Saudi Arabia

Corresponding authors
Nofe Alganmi,
nalghanimi@kau.edu.sa
Arwa Bashanfar,
abashanfar0002@stu.kau.edu.sa

## ABSTRACT

Breast and ovarian cancers are prevalent worldwide, with genetic factors such as BRCA1 and BRCA2 mutations playing a significant role. However, not all patients carry these mutations, making it challenging to identify risk factors. Researchers have turned to whole exome sequencing (WES) as a tool to identify genetic risk factors in BRCA-negative women. WES allows the sequencing of all protein-coding regions of an individual's genome, providing a comprehensive analysis that surpasses traditional gene-by-gene sequencing methods. This technology offers efficiency, cost-effectiveness and the potential to identify new genetic variants contributing to the susceptibility to the diseases. Interpreting WES data for disease-causing variants is challenging due to its complex nature. Machine learning techniques can uncover hidden genetic-variant patterns associated with cancer susceptibility. In this study, we used the extreme gradient boosting (XGBoost) and random forest (RF) algorithms to identify BRCA-related cancer high-risk genes specifically in the Saudi population. The experimental results exposed that the RF method scored superior performance with an accuracy of 88.16% and an area under the receiver-operator characteristic curve of 0.95. Using bioinformatics analysis tools, we explored the top features of the high-accuracy machine learning model that we built to enhance our knowledge of genetic interactions and find complex genetic patterns connected to the development of BRCA-related cancers. We were able to identify the significance of HLA gene variations in these WES datasets for BRCA-related patients. We find that immune response mechanisms play a major role in the development of BRCA-related cancer. It specifically highlights genes associated with antigen processing and presentation, such as HLA-B, HLA-A and HLA-DRB1 and their possible effects on tumour progression and immune evasion. In summary, by utilizing machine learning

approaches, we have the potential to aid in the development of precision medicine approaches for early detection and personalized treatment strategies.

## INTRODUCTION

Approximately 19.3 million people were diagnosed with cancer in 2020, causing 10 million deaths, according to GLOBOCAN 2020 (*Sung et al., 2021*). Breast cancer—a BRCA-related cancer—is the most-diagnosed cancer type in the world. Breast cancer was diagnosed in 2.3 million cases, which represented 11.7%, and there were 68,496 deaths, representing 6.9% of all cancer deaths. In total, 17.8% and 3%, breast cancer and ovarian cancers, respectively, are the most common types of cancer among Saudi nationality (*National Cancer Center of the Saudi Health Council, 2024*). While genetic factors are known to play a significant role in their development, specifically mutations in the breast cancer 1 (BRCA1) and breast cancer 2 (BRCA2) genes, it is important to note that not all breast and ovarian cancer patients carry these specific mutations. This poses challenges when identifying genetic risk factors in these patients (*Kurian et al., 2011*). The BRCA1 and BRCA2 genes are well-known tumor suppressor genes that play a crucial role in maintaining the stability of the genome. Mutations in these genes significantly increase the risk of developing hereditary breast and ovarian cancers (*Petrucelli, Daly & Pal, 2022*). The prevalence of BRCA mutations in breast and ovarian cancer patients can vary among different populations. Studies have estimated that only approximately around 5% to 10% of breast and ovarian cancer patients have BRCA mutations (*Godet & Gilkes, 2017*). This indicates that a substantial number of patients who test negative for these mutations may still have an unidentified genetic risk factor contributing to their condition. There are several limitations of BRCA testing in identifying genetic risk factors in BRCA-negative patients. One limitation is the low occurrence of BRCA mutations in certain breast and ovarian cancer patients, leading to false-negative results. Moreover, BRCA testing only targets specific genes and mutations and does not analyze the entire genome for potential genetic risk factors. This means that other genetic variants that may contribute to breast and ovarian cancer susceptibility may be missed. Identifying genetic risk factors in patients who test negative for BRCA mutations requires alternative approaches. One such approach is whole exome sequencing (WES), which can identify genetic variants associated with breast and ovarian cancer susceptibility beyond BRCA1 and BRCA2 mutations. WES is an advanced DNA analysis technique that allows the sequencing of all protein-coding regions, known as exons, in one comprehensive assay. This method offers several advantages compared to other sequencing techniques such as whole-genome sequencing (WGS) and targeted gene panels (*Bartha & Győrffy, 2019*). One significant advantage of WES is its efficiency and cost effectiveness (*Suwinski et al., 2019*). WES also offers the advantage of

identifying new genetic variants that could potentially contribute the susceptibility to the disease. By analyzing genes beyond those already known to cause disease, WES provides a more thorough examination of the genetic factors underlying various conditions (*Rabbani, Tekin & Mahdieh, 2014*).

Findings from studies using WES in BRCA-negative breast and ovarian cancer have identified specific genes and variants associated with the susceptibility to breast and ovarian cancer. These studies have provided valuable insights into the genetic factors underlying these cancers, potentially informing personalised treatment and risk management approaches. For instance, a recent study by *Felicio et al. (2021)* used WES analysis to identify new genes linked to the predisposition to breast and ovarian cancer. The researchers examined germline variants in cancer-related genes and conducted bioinformatic analyses to pinpoint potential genetic risk factors. Notably, they discovered a variant called c.149T > G in the FAN1 gene, present in two unrelated families, with a loss of heterozygosity observed in one family's tumor tissue. This finding suggests that FAN1 may be a promising candidate associated with hereditary breast and ovarian cancer susceptibility (*Felicio et al., 2021*). In a recent study by *Lee et al. (2022)*, the researchers used WES and case-control analyses to investigate genetic variants that might be linked to breast cancer susceptibility in individuals without BRCA mutations. Their findings uncovered new and potentially important genetic variants that could contribute to the risk of breast cancer in this particular population. Among the candidate genes identified, one of note was MUC16, which has previously been associated with increased susceptibility to ovarian cancer (*Lee et al., 2022*). Another study by *BenAyed-Guerfali et al. (2022)* focused on BRCA-negative Tunisian patients at high risk of hereditary breast/ovarian cancer. Using WES, the study aimed to identify other genes that may contribute to susceptibility. Their findings shed light on potential genetic risk factors beyond BRCA mutations, highlighting candidate genes such as RAD51C, which have been previously linked to the susceptibility to breast and ovarian cancer (*BenAyed-Guerfali et al., 2022*). Furthermore, a recent study conducted by *Grasel et al. (2020)* used WES to identify genetic variants that may contribute to an increased risk of breast cancer in individuals from families with a history of the disease who tested negative for BRCA mutations. The study revealed potential candidate genes, one of which is the ATM gene, previously linked to the susceptibility to breast cancer.

Conversely, the process of interpreting WES data and finding the disease-causing variants among thousands of variants remains a challenge due to the characteristics of these data (*Huang et al., 2022*). Thus, using machine learning (ML) with these kinds of data supports humans in dealing with these large and complex data (*Fan et al., 2022*).

Many studies profit from ML algorithms using WES for different diseases. For example, in *Trakadis et al. (2019)*, they applied extreme gradient boosting (XGBoost), random forest (RF), L1.Logistic and a support vector machine (SVM) on WES data to identify the high-risk genes for the schizophrenia (SCZ) disease. XGBoost yielded ideal results with an accuracy equal to 85.7%, a specificity equal to 86.6%, a sensitivity equal to 84.9% and an area under the receiver operator characteristic (AUC) equal to 0.95. Moreover, they analyzed the top 50 genes of the algorithms that are associated with SCZ using

bioinformatic resources. In addition, in *Hooshmand (2020)*, they applied a naive Bayesian algorithm on 1,091 ovarian cancer samples from The Cancer Genome Atlas (TCGA) research network and 179 healthy people from the Genotype-Tissue Expression (GTEx) project to distinguish between cancerous and noncancerous cells. They yielded amazing results with 100% accuracy, 100% specificity, 100% sensitivity and an AUC equal to one. Additionally, in *Wadapurkar et al. (2023)*, they applied five supervised machine learning algorithms-SVM, decision tree (DT), RF, Naive Bayes and XGBoost - to identify variants that are associated with ovarian cancer. XGBoost yielded 94.64% in accuracy and 0.97 as the AUC. RF and XGBoost are efficient machine learning methods because they work well for problems with high dimensions and they generally allow nonlinear relationships between the features (ref).

In conclusion, it is crucial to continue exploring genetic risk factors for breast and ovarian cancers in individuals who do not have BRCA1 and BRCA2 mutations. While these mutations are significant risk factors, a considerable number of patients lack them, underscoring the need to identify other robust biomarkers. Machine learning techniques for analyzing WES data have emerged as a powerful tool for uncovering hidden genetic variant patterns associated with the susceptibility to cancer beyond the established BRCA genes.

## MATERIALS AND METHODS

The model, which contains three stages-data pre-processing, machine learning and pathway analysis-is shown in Fig. 1. The complete source code and data files are available on Zenodo at https://zenodo.org/records/10720553, https://doi.org/10.5281/zenodo.10926612.

### Exome data set

We utilized WES data obtained from the Center of Excellence in Genomic Medicine Research (CEGMR) at King Abdulaziz University. The dataset includes 76 women who were diagnosed with BRCA-related cancer but tested negative for known risk variants (BRCA genes). Ethical approval for this study was obtained under IRB No. 32-CEGMR-Bioeth-2021, and written informed consent was obtained from all participants. Among the participants, 21 individuals had a positive BRCA genetic status, while 55 individuals had a negative BRCA genetic status. The data files were provided in the VCF file, containing a total of 249,430 germline short variants (single nucleotide polymorphisms (SNP) and indel) calls. Figure 2 illustrates the screenshot of the VCF file.

### VCF file pre-processing

Raw whole-exome sequencing data were aligned with the human reference genome GRCh38 and then called using GATK's (v.4.1.2) HaplotypeCaller. After that, BCFtools V.1.13.1 was used to merge all VCF files into one cohort VCF. In the merged VCF file, all SNPs are biallelic SNPs that have been filtered to remove calls with a depth of less than 20. The data were annotated by ANNOVAR (*Wang, 2010*) using the reference genome hg38/GRCh38.

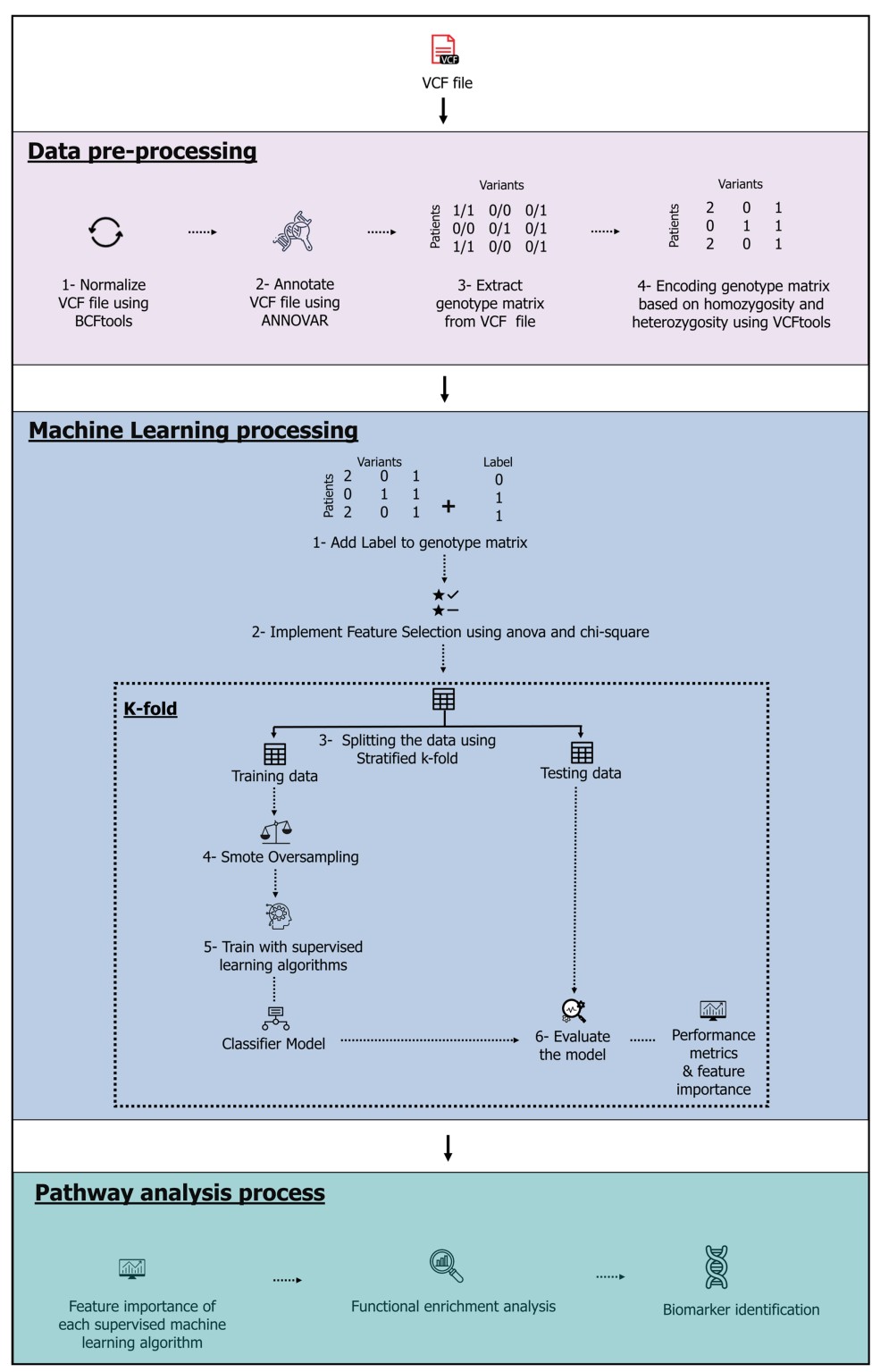

**Figure 1 The machine learning pipeline.** Image source credits: Feature selection icons created by herikus-Flaticon. Data table icon, Designed by Freepik. Machine learning icon, Designed by Freepik. Magnifying glass icons created by Freepik-Flaticon. Classifier model icon, Designed by Freepik. Benchmark icons created by srip-Flaticon. Genes icons created by Smashicons-Flaticon.

|  | CHROM | POS | ID | REF | ALT_1 | ALT_2 | ALT_3 | QUAL |
|---|---|---|---|---|---|---|---|---|
| 0 | chr1 | 13380 | rs571093408 | C | G | NaN | NaN | 2064.10009 |
| 1 | chr1 | 13417 | rs777038595 | C | CGAGA | NaN | NaN | 17749.19921 |
| 2 | chr1 | 13418 | rs75175547 | G | A | NaN | NaN | 43.90000 |
| 3 | chr1 | 13445 | rs558318514 | C | G | NaN | NaN | 513.20001 |
| 4 | chr1 | 13550 | rs554008981 | G | A | NaN | NaN | 139.10000 |
| ... | ... | ... | ... | ... | ... | ... | ... | . |
| 52817 | chrM | 15812 | rs200336777 | G | A | NaN | NaN | 1284.09997 |
| 52818 | chrM | 15884 | rs527236195 | G | C | NaN | NaN | 306.29998 |
| 52819 | chr22_KI270879v1_alt | 271020 | rs2266637 | C | T | NaN | NaN | 210.00000 |
| 52820 | chr22_KI270879v1_alt | 277188 | rs8140585 | C | T | NaN | NaN | 142.50000 |
| 52821 | chr22_KI270879v1_alt | 278394 | rs56106137 | G | A | NaN | NaN | 307.60000 |

249430 rows × 9 columns

## VCF file

**Figure 2 The snapshot of the dataset (VCF file).**

The last step in this phase is encoding the genotype matrix to be ready to use in the machine learning phase. The encoding step was done using VCFtools (*Danecek et al., 2011*) Version 0.1.16-3 through the '012 matrix' command (*Figueiredo de Sá et al., 2019*; *Magi et al., 2015*; *Carvalho et al., 2020*). VCFtools provides simple tools for working with genetic variation data stored in VCF files. The encoding accomplished was as follows:

- 0/0 denotes the reference homozygote that is coded as 0.
- 0/1 and 1/0 denote the heterozygote that is coded as 1.
- 1/1 denotes the alternative homozygote that is coded as 2.
- All missing values are coded as −1.

Figure 3 illustrates the final data used in the machine learning process.

## Machine learning process

We used random forest (RF) and XGBoost supervised machine learning algorithms with the labeled genotype matrix for the 76 individual datasets to identify features or genes that might correlate with BRCA-related risk factors. The following subsections discuss the feature selection process and machine learning in detail.

## Feature selection process

As a preliminary step, we employed ANOVA and chi-square feature selection techniques in order to prevent overfitting. Using the top 5,000 and 7,000 variants of each technique, we reduced the number of variants in the genotype matrix. Thus, after applying feature selection techniques, we obtained the top 500 and 700 variants for each feature selection method (ANOVA and chi-square).

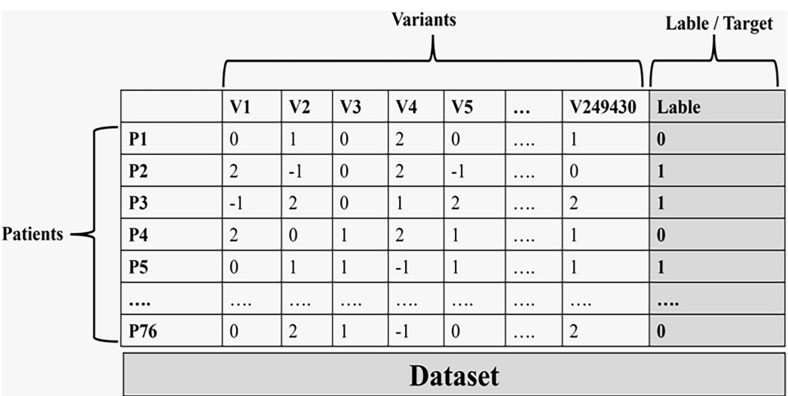

**Figure 3** The final data set that is used in the machine learning process.

## Machine learning process

For splitting data into training and testing datasets, we applied stratified k-fold cross-validation with the k equal to four (*Bukhari et al., 2022*; *Bukhari, Webber & Mehbodniya, 2022*). For the training data, the Synthetic Minority Oversampling Technique (SMOTE) was used to avoid problems of imbalanced data using the imbalanced-learn package Version 0.10.1. To evaluate the classifier model, the testing datasets were used. Table 1 shows an illustration of the parameters of the supervised machine learning algorithms which that chosen through GridSearchCV.

## Molecular pathway and network analysis
### *Molecular pathway analysis*

The machine learning models were used to identify the top 100 significant variants through feature importance. Gene ontology (GO) analysis and Kyoto Encyclopedia of Genes and Genomes (KEGG) pathway analysis were conducted on the significant top variants and their corresponding genes using DAVID and R to explore the potential mechanisms and risk factors associated with BRCA-related cancer, including both BRCA-positive and BRCA-negative patients. The statistically significant enriched terms were considered for an adjusted *P*-value less than 0.01.

## RESULTS AND DISCUSSION
### Machine learning process

In order to identify the most important variants and genes related to ovarian cancer and breast cancer in the Saudi population, a prediction model was trained using machine learning techniques. Specifically, we employed the XGBoost and RF algorithms. The performance of these models was evaluated using various metrics, including accuracy, precision, recall, F1 score and AUC.

Table 2 presents the results of the feature selection techniques used to improve the performance of the machine learning algorithms. The feature selection technique ANOVA, using the top 5,000 variants, achieved an accuracy of 78.95% for XGBoost and 88.16% for RF. On the other hand, chi-squared feature selection demonstrated the best

**Table 1 List of the machine learning algorithms' parameters with optimized values.**

| Machine learning algorithm | Parameter name | Parameter value |
| --- | --- | --- |
| XGBoost | Loss function | log_oss |
| | Number of boosting stages | 100 |
| | Function to measure the quality of a split | friedman_mse |
| | Minimum number of samples required to split | 2 |
| | Minimum number of samples required to be at a leaf node | 1 |
| RF | Criterion of trees | gini |
| | Number of trees in the forest | 100 |
| | Minimum number of samples required to split | 2 |
| | Minimum number of samples required to be at a leaf node | 1 |
| | Number of features for the best split | sqrt |

**Table 2 Performance metrics among supervised machine learning algorithms with ANOVA and chi-squared feature selection (5,000 features).**

| Model\Metric | | Accuracy | Precision | Recall | F1 score | AUC |
| --- | --- | --- | --- | --- | --- | --- |
| ANOVA | XGBoost | 78.95% | 72.50% | 46.67% | 52.63% | 0.79 |
| | RF | 88.16% | 100% | 55.83% | 68.56% | 1.0 |
| Chi-squared | XGBoost | 82.89% | 81.67% | 52.5% | 64.64% | 0.81 |
| | RF | 88.16% | 95% | 60.83% | 71.30% | 0.95 |

performance in this study, particularly in combination with RF. Figure 4 shows the ROC curve for the best-performing algorithm. The accuracy of XGBoost was 82.89%, while RF achieved an accuracy of 88.16%. Additionally, the F1 score improved from 52.63% to 64.64% with XGBoost and from 68.56% to 71.30% with RF.

## Gene ontology analysis

The analysis revealed an enrichment in the biological processes regarding the immune response, such as antigen processing and presentation, T cell-mediated cytotoxicity and immune cell differentiation (Fig. 5). These findings highlighted the crucial role of immune mechanisms in the evolution of BRCA-related cancer. Additionally, genes associated with MHC class I and II molecules, including HLA-B, HLA-A and HLA-DRB1, were identified, emphasizing the significance of antigen presentation and immune recognition in this cancer (*Liu et al., 2021*). Disruptions in these processes may facilitate immune evasion and contribute to tumour advancement. Moreover, the enrichment of the genes involved in protein hetero-tetramerization and cell adhesion suggested the potential involvement of cellular interactions and communication in BRCA-related cancers (*Godet & Gilkes, 2017*). Alterations of these aspects may disrupt cell-cell adhesion, promoting cancer cell metastasis and invasiveness. Unexpectedly, the analysis also revealed the enrichment of genes involved in tissue-specific processes such as muscle organ development, heart development and spermatogenesis. This suggested potential correlations between these

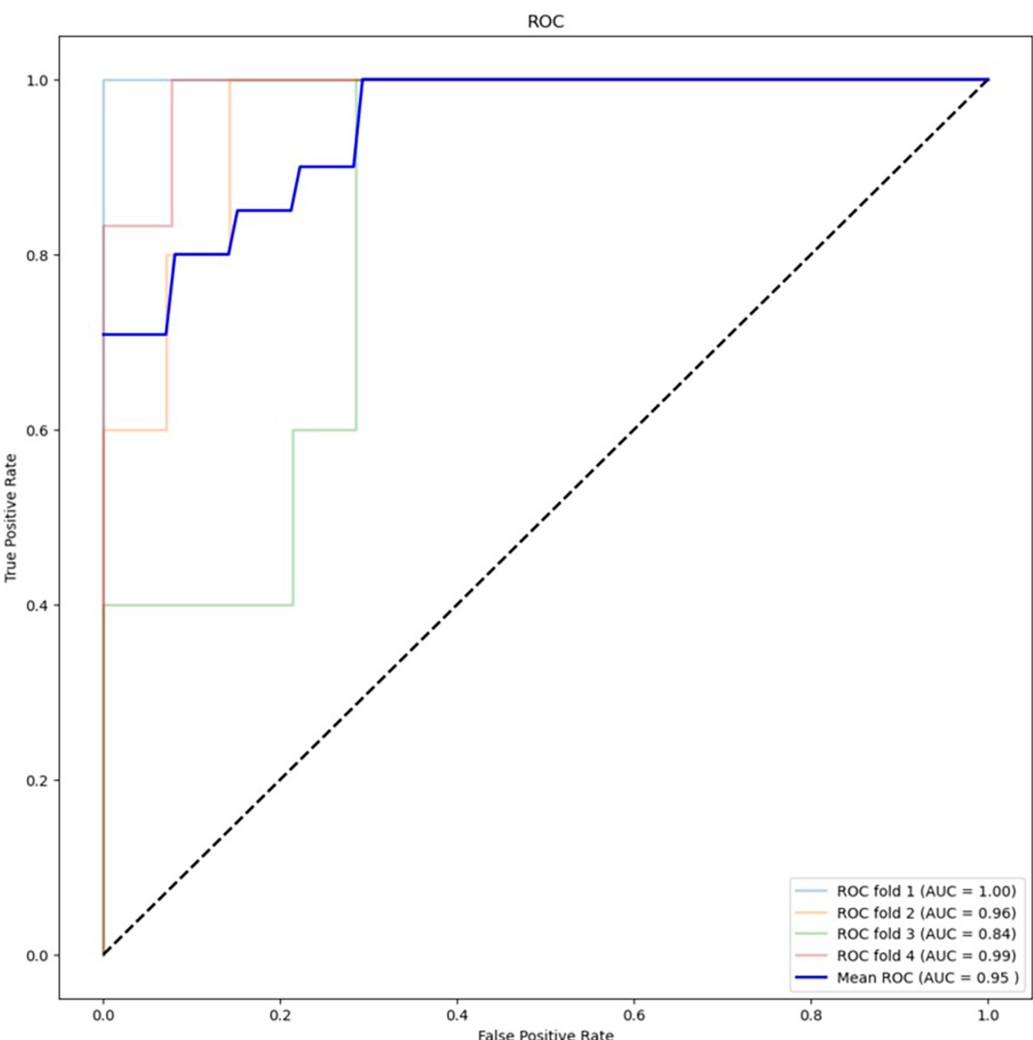

**Figure 4 ROC curve for best performing algorithm: RF with chi-square (5,000 variants).**

processes and BRCA-related cancers, indicating the possible role of tissue-specific dysfunctions in cancer susceptibility or phenotype expression (*Zhang & Li, 2018*). Table 3 shows the key findings (genes) and their pivotal functions.

## KEGG pathway analysis

The KEGG pathway analysis using DAVID (*Sherman et al., 2022*; *Huang, Sherman & Lempicki, 2009*) identified several pathways of potential relevance to BRCA-related cancers. The enrichment of pathways associated with the immune response, immunological diseases, natural killer cell-mediated cytotoxicity and allograft rejection further emphasises the importance of immune cell interactions in these cancers. Additionally, pathways related to hormone signaling, including the estrogen-signalling pathway, were enriched, suggesting the influence of estrogen signalling even in BRCA-negative patients (*Yin et al., 2020*).

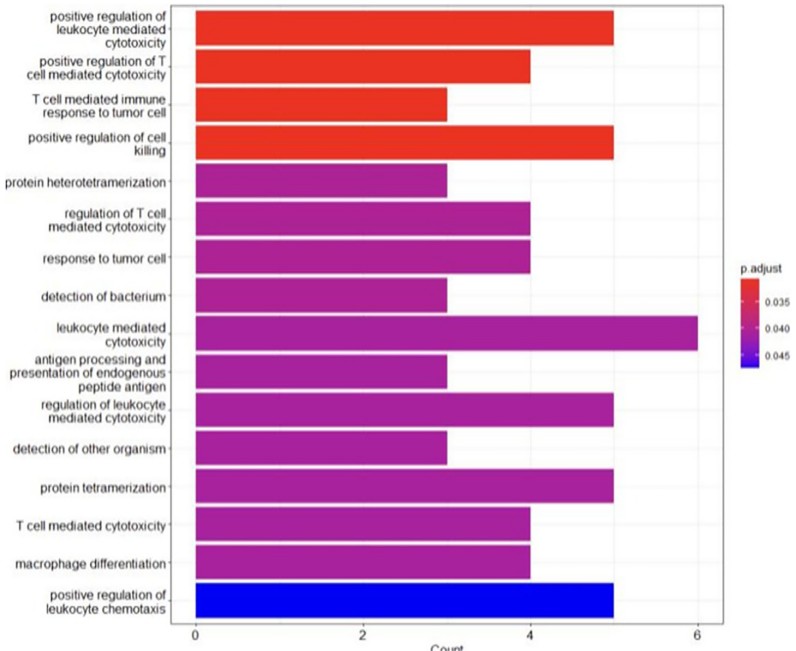

**Figure 5** **The GO analysis conducted in R reveals significant biological processes linked to the genes of interest.**

**Table 3** **Table encapsulating the regarding genes and their pivotal functions.**

| Gene | Process involved in | Description/Pivotal function |
| --- | --- | --- |
| HLA-A | Antigen processing and presentation | Crucial for immune response, presents peptide antigens to T cells |
| HLA-B | Graft-versus-host disease, allograft rejection | Plays a key role in immune system recognition, impacting graft compatibility |
| HLA-DRB1 | Antigen processing and presentation | Involved in presenting antigens to immune cells, crucial to the immune response |
| KIR2DL1 | Natural killer cell-mediated cytotoxicity | Regulates natural killer cell activity, important for immune surveillance |
| KIR2DL3 | Natural killer cell-mediated cytotoxicity | Influences the activity of natural killer cells, key in immune defense mechanisms |
| KIR2DS1 | Natural killer cell-mediated cytotoxicity | Affects natural killer cell function, important in immune system's response |
| KIR2DS5 | Natural killer cell-mediated cytotoxicity | Modulates the activity of NK cells, playing a role in immune response |
| GHR | Estrogen signalling pathway | While primarily known for growth hormone receptor, may interact with estrogen signaling |
| SHC1 | Estrogen signalling pathway | Involved in signal transduction, potentially linking to estrogen pathways |
| SIRT1 | Estrogen signalling pathway | Known for roles in aging and metabolism, may have links to estrogen signaling |

These findings highlighted the complex interplay of factors, including immune-related processes, hormone signalling and natural killer cell-mediated cytotoxicity, as potential risk factors for BRCA-related cancers, extending beyond BRCA mutations' cell targeting. For example, genes involved in antigen processing and presentation, TAP binding and T cell receptor binding were implicated, indicating their potential roles in immune recognition and cancer (*Reeves & James, 2017*). The enrichment of the graft-*vs.*-host disease pathway suggests that immune dysregulation may contribute to BRCA-related

**Table 4 The KEGG pathways associated with the analysed genes, highlighting their involvement in specific biological pathways and networks.**

| Category | Term | RT | Count | % | *P*-value | Benjamini |
|---|---|---|---|---|---|---|
| KEGG_PATHWAY | Antigen processing and presentation | RT | 8 | 5.7 | 6.9E−7 | 1.1E−4 |
| KEGG_PATHWAY | Graft-*vs*-host disease | RT | 6 | 4.3 | 7.5E−6 | 5.7E−4 |
| KEGG_PATHWAY | Natural killer cell-mediated cytotoxicity | RT | 6 | 4.3 | 1.5E−3 | 7.4E−2 |
| KEGG_PATHWAY | Estrogen signalling pathway | RT | 5 | 3.5 | 1.3E−2 | 4.9E−1 |
| KEGG_PATHWAY | Allograft rejection | RT | 3 | 2.1 | 2.6E−2 | 7.9E−1 |
| KEGG_PATHWAY | Type I diabetes mellitus | RT | 3 | 2.1 | 3.3E−2 | 8.3E−1 |
| KEGG_PATHWAY | Autoimmune thyroid disease | RT | 3 | 2.1 | 4.8E−2 | 1.0E0 |
| KEGG_PATHWAY | Human T-cell leukemia virus 1 infection | RT | 5 | 3.5 | 5.9E−2 | 1.0E0 |
| KEGG_PATHWAY | Viral myocarditis | RT | 3 | 2.1 | 6.0E−2 | 1.0E0 |

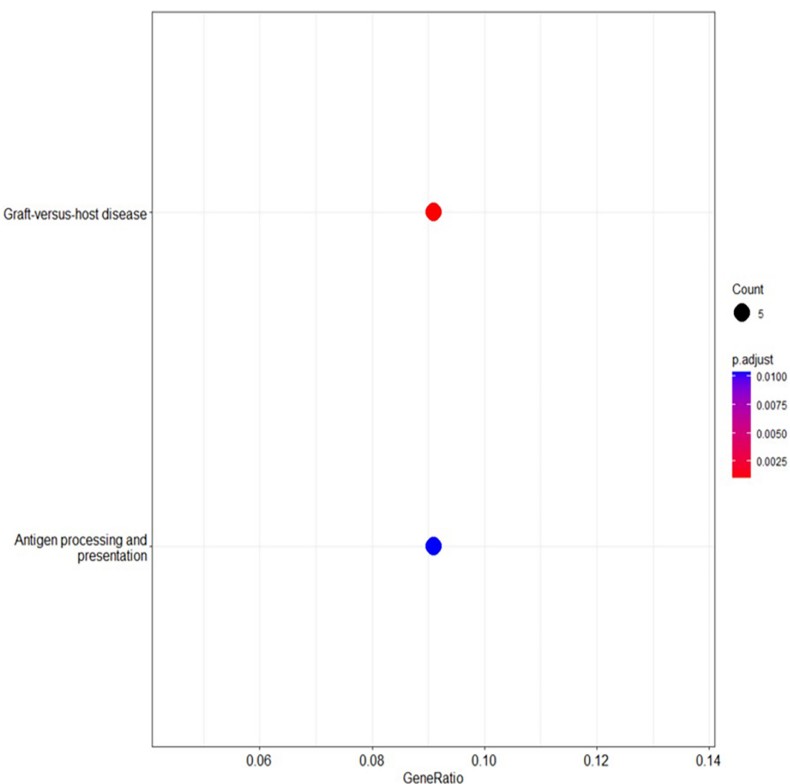

**Figure 6 The enriched KEGG pathways analysed using R linked to the selected genes, providing insights into their functional relationships and potential roles in biological processes.**

cancer susceptibility (Table 4). Additionally, metalloendopeptidase activity and calcium channel activity were identified as potential mechanisms, pointing to the involvement of extracellular matrix remodeling, cell migration and calcium signalling in cancer progression (*Chen et al., 2013*).

The analysis of the KEGG pathway using the KEGG Mapper, as shown in Figs. 6 and 7 (*Kanehisa & Sato, 2020*; *Kanehisa, Sato & Kawashima, 2022*; *Kanehisa & Goto, 2000*;

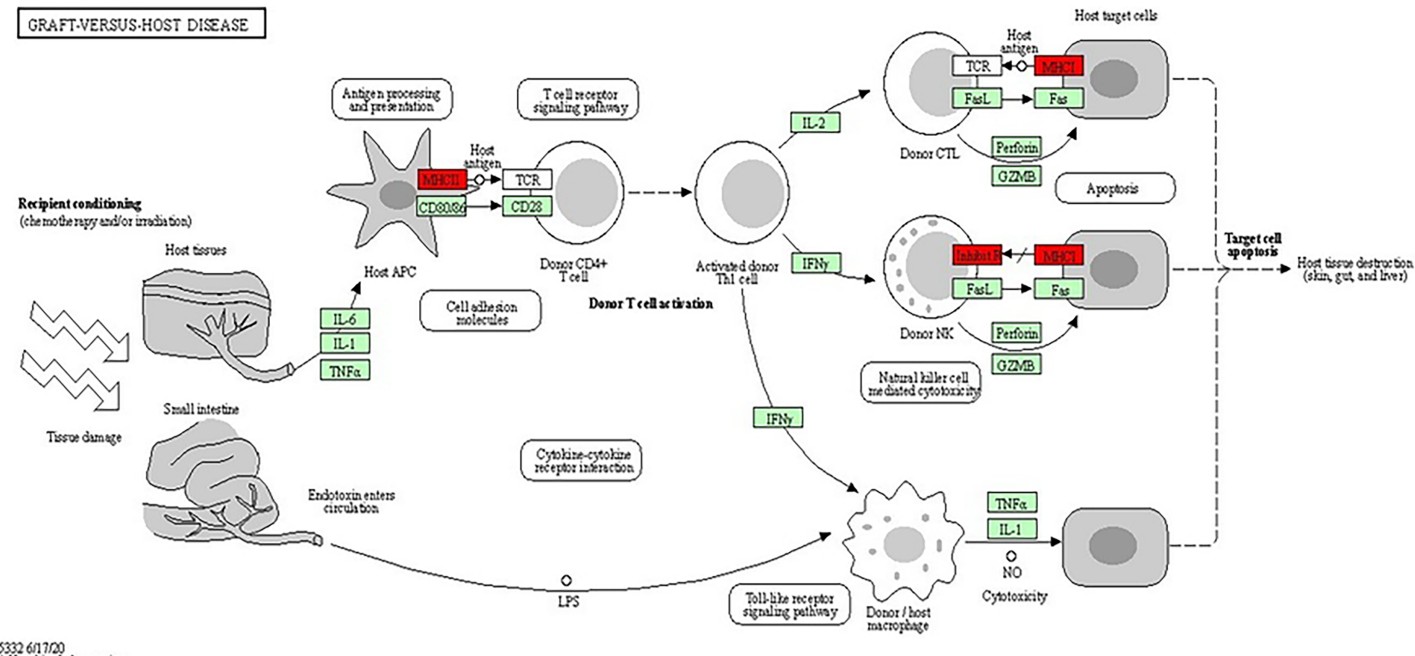

**Figure 7 KEGG pathway diagram for graft-*vs.*-host disease depicting the complex interplay of cellular and molecular interactions involved in the pathogenesis of graft-*vs.*-host disease (GVHD).** Key components include donor T cells, antigen-presenting cells (APCs) and various cytokines. The detailed interactions between these components highlighted the roles of immune response and inflammation in GVHD progression (Image source credit: Kanehisa Laboratories).              

*Kanehisa et al., 2023*) revealed only two significant pathways with implications for BRCA-related cancers: the graft-*vs.*-host disease pathway and the antigen processing and presentation pathway. The 'graft-*vs.*-host disease pathway' pathway was identified as an enriched pathway in the analysis. This pathway is typically associated with complications that can occur after haematopoietic stem cell transplantation (HSCT), where immune cells from the donor (graft) attack the recipient's (host) tissues (*Zeiser & Blazar, 2017*). The involvement of this pathway in BRCA-related cancers suggests a potential role for immune dysregulation in contributing the susceptibility to cancer. The presence of genes associated with this pathway pointed to the importance of immune-related factors beyond BRCA mutations.

The antigen processing and presentation pathway, a fundamental process for immune recognition and elimination of pathogens or abnormal cells, including cancer cells, was also enriched. This pathway (shown in Fig. 8) involves the presentation of antigens to immune cells, initiating an immune response. Antigens are processed and presented on the cell's surface through two major pathways: the major histocompatibility complex class I (MHCI) pathway and the major histocompatibility complex class II (MHCII) pathway (*Wieczorek et al., 2017*). In the MHCI pathway, intracellular proteins, such as viral or tumour antigens, are broken down into small peptide fragments within the cytosol. These peptides are then transported into the endoplasmic reticulum (ER), where they bind to MHCI molecules. The MHCI-peptide complexes are subsequently presented on the cell's

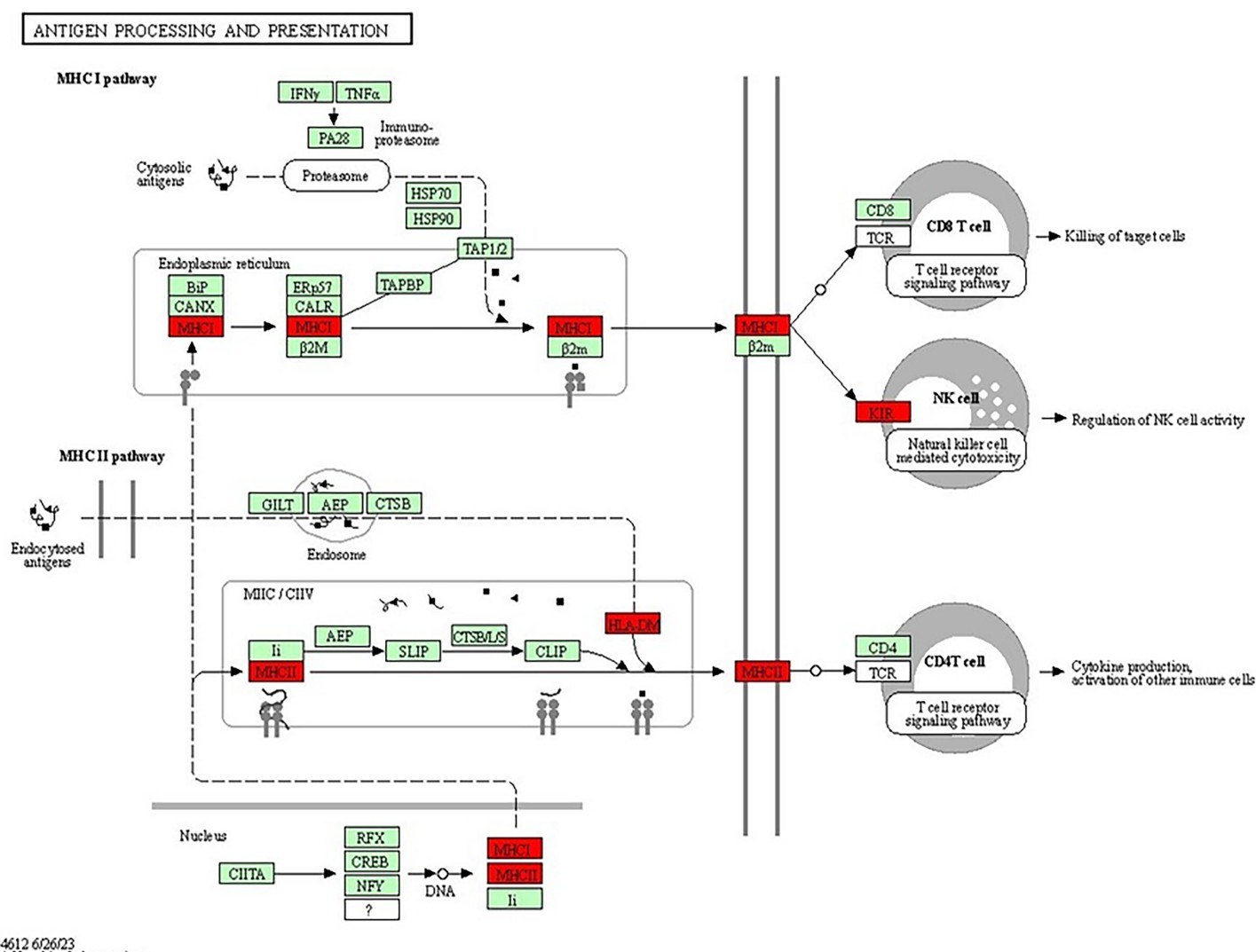

**Figure 8 KEGG pathway diagram for antigen processing and presentation illustrating the central processes involved in antigen processing and presentation, which are essential for immune system function.** The diagram focuses on the breakdown of pathogens into peptide fragments, their subsequent association with MHC molecules and their presentation to T cells, thereby triggering an immune response (Image source credit: Kanehisa Laboratories).

surface, allowing cytotoxic T cells to recognise and eliminate infected or malignant cells. The presence of genes associated with MHCI-mediated antigen presentation, including HLA-B, HLA-A and HLA-DRB1, suggested their potential roles in BRCA-related cancers. Dysregulation or abnormalities in this pathway may impact the immune system's ability to recognise and combat cancer cells, independent of BRCA mutations. The MHCII pathway involves the presentation of antigens derived from extracellular sources, such as bacteria or proteins released from dying cells (*Abualrous, Sticht & Freund, 2021*). Antigen-presenting cells (APCs), including dendritic cells, macrophages and B cells, take up the extracellular antigens. Within the APCs, the antigens are processed into peptide fragments and loaded

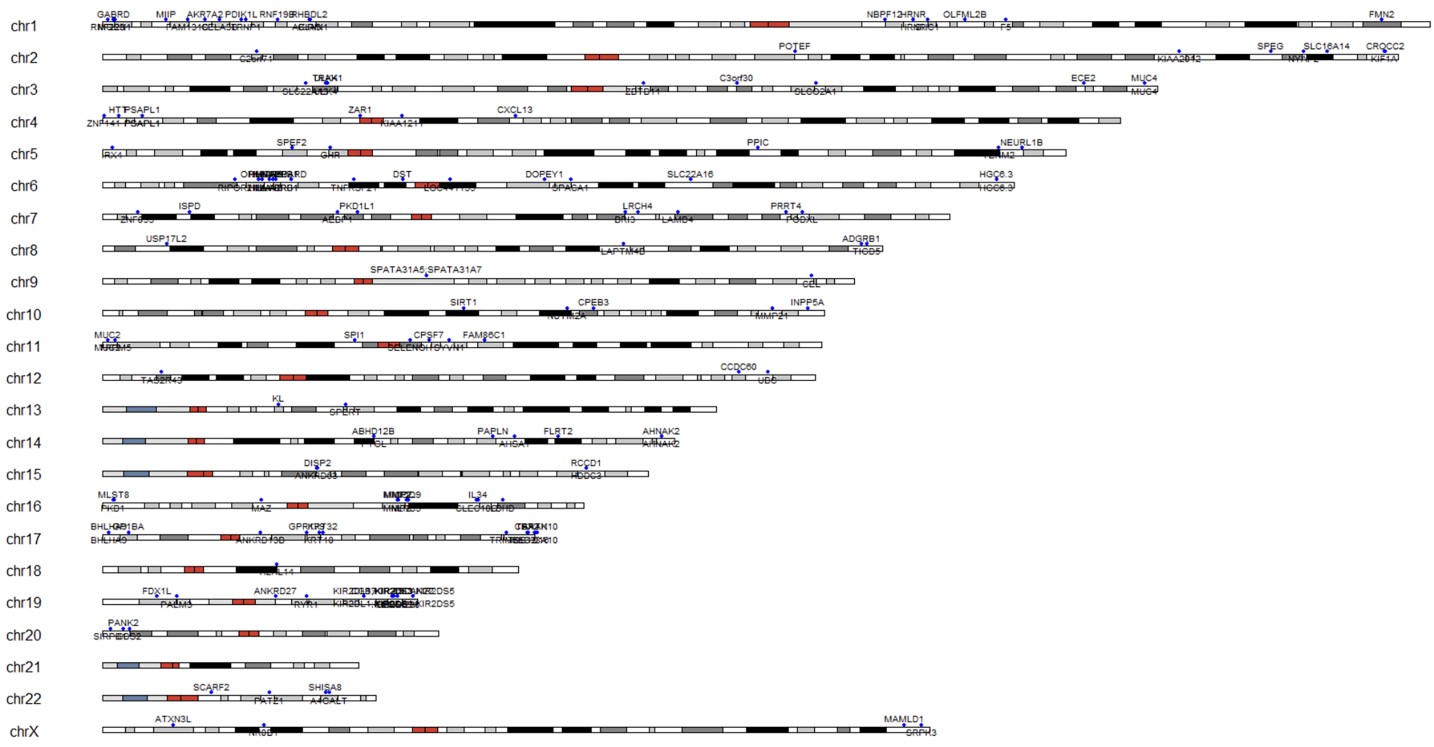

**Figure 9 Karyotype with SNP locations represented by blue points.** Each blue point is labeled with the corresponding gene name, indicating the genes associated with the identified variants. The figure offers a comprehensive view of the genetic variations within the karyotype and their specific gene associations.

onto MHCII molecules. The MHCII-peptide complexes are then displayed on the cell's surface, activating helper T cells to coordinate the immune response. The identification of genes associated with MHCII-mediated antigen presentation, such as HLA-DM, further emphasizes the involvement of immune recognition and immune response mechanisms in BRCA-related cancers. The dysregulation of these pathways and abnormalities in immune recognition and response may contribute to the development and progression of BRCA-related cancers. Studies have shown that PALB2, ATM, CHEK2 and TP53, among others, have also been associated with the increased risk of breast cancer (*Easton et al., 2015*; *Gracia-Aznarez et al., 2013*). For example, TP53 mutations are known to play a significant role in Li-Fraumeni syndrome, which elevates the risk of breast cancer as well as other types of cancer (*Gracia-Aznarez et al., 2013*).

## Variant analysis in genes associated with cancer-related pathways

The study sought to identify additional genetic variants associated with cancer susceptibility in patients who are BRCA1/2 mutation-negative but have BRCA-related cancers. The analysis examined genetic variants in genes known to be involved in cancer-related pathways, beyond BRCA1/BRCA2 mutations. These variants may represent other genetic risk factors for BRCA-related cancers (see Fig. 9).

The search for cancer risk-associated genes was accomplished by focusing on variants located especially in known cancer-related pathways that result in non-synonymous mutations, potentially affecting protein function, and are more prevalent in cancer patients compared to BRCA-negative patients. The role of non-BRCA genes in cancer-related pathways is increasingly being recognized (*Evans et al., 2021*). These genes, which include PALB2, ATM and CHEK2 among others, are known to interact with BRCA1 and BRCA2 in DNA repair pathways and mutations in these genes have been associated with a moderate increase in the risk of breast and ovarian cancer (*Couch et al., 2017*; *Cybulski et al., 2015*; *Easton et al., 2015*). Non-synonymous mutations, which result in changes to the protein sequence, have a higher likelihood of affecting protein functioning compared to synonymous mutations (*Miosge et al., 2015*). These mutations can alter protein activity, stability or interaction with other proteins, leading to the dysregulation of cellular pathways. In the context of BRCA-related cancers in patients who are BRCA negative, non-synonymous mutations may represent additional genetic risk factors.

By observing non-synonymous mutations in genes known to be involved in DNA repair or cell cycle control, we identified potential disruptions in normal cellular functions that could contribute to an increased risk of cancer. For example, a non-synonymous variant in the RNF223 gene, which changes the proline at position 231 to arginine (P231R), may impact the protein's function. BRCA-related cancer patients exhibit a diverse range of genetic variants, emphasising the polygenic nature of cancer susceptibility (*Vihinen, 2022*). These variants occur in multiple genes, some of which are already known to be associated with cancer, while others may represent novel candidate susceptibility genes. For instance, the dataset revealed a nonsynonymous variant in the RNF223 gene, which encodes an E3 ubiquitin-protein ligase (*Feng et al., 2021*). Disruptions in such genes can interfere with the protein degradation process, potentially leading to the accumulation of harmful proteins that promote cancer. Additionally, the GABRD gene encodes a subunit of the gamma-aminobutyric acid (GABA) receptor (*Wu et al., 2023*). Perturbations in GABA signalling have been implicated in various cancers, suggesting that alterations in this pathway could contribute to the development of cancer.

Additionally, several variants were found in non-coding regions or led to synonymous changes (no change in the encoded amino acid). While these variants do not alter protein sequences, they can still influence gene regulation and expression, thereby potentially contributing to cancer susceptibility (*Sharma et al., 2019*). These regions, once considered 'junk DNA', are now recognized as crucial regulatory regions capable of influencing gene expression. Variants in these regions can impact the binding of transcription factors or other regulatory molecules, leading to changes in the expression levels of associated genes. Such regulatory alterations can profoundly affect cellular processes and potentially contribute to the development or progression of cancer. They can influence mRNA stability, splicing and translation, thereby impacting protein levels within the cell. For instance, the analysis identified a synonymous variant in the FAM131C gene. Although this variant does not change the protein sequence it could still influence the expression or function of FAM131C, which may be relevant to cancer (*Gotea et al., 2015*). Taking this into consideration, further research is required to validate these potential risk factors,

investigate their interactions with each other and determine their interplay with non-genetic factors. Such endeavors will contribute to a more comprehensive understanding the cancer risk in BRCA-negative patients.

## CONCLUSION

In this study, we used WES samples from 76 women with BRCA-related cancer who tested negative for BRCA gene mutations. We present a machine-learning approach to detect the high-risk genes in the Saudi population. The RF algorithm achieves 88% accuracy and an AUC of 0.95 on the testing dataset. The learned model has been analysed to extract the features as variants to be studied. The analysis of the data and the results indicate that changes in HLA genes affecting the immune system are crucial to the development of cancers related to BRCA. Thus, understanding the interplay between immune dysregulation and BRCA-related cancers could provide valuable insights for the development of immunotherapeutic strategies and improved treatment outcomes. Further research is required to validate these potential risk factors, investigate their interactions with each other and determine their interplay with non-genetic factors. Such endeavours will contribute to a more comprehensive understanding of cancer risk in BRCA-negative patients.

## ACKNOWLEDGEMENTS

The authors extend their appreciation to Dr. Robert Hoehndorf, Dr. Ashraf Dallol and Dr. Heba Alkatabi for their interesting discussions on this work.

### Funding
This work was supported by the Deputyship for Research and Innovation, Ministry of Education in Saudi Arabia through the project number IFPNC-008-141-2020 and King Abdulaziz University, DSR, Jeddah, Saudi Arabia. The funders had no role in study design, data collection and analysis, decision to publish, or preparation of the manuscript.

### Grant Disclosures
The following grant information was disclosed by the authors:
Deputyship for Research and Innovation, Ministry of Education in Saudi Arabia: IFPNC-008-141-2020.
King Abdulaziz University, DSR, Jeddah, Saudi Arabia.

### Competing Interests
Sajjad Karim is an Academic Editor for PeerJ.

### Author Contributions
- Nofe Alganmi conceived and designed the experiments, performed the experiments, analyzed the data, performed the computation work, prepared figures and/or tables, authored or reviewed drafts of the article, and approved the final draft.

- Arwa Bashanfar conceived and designed the experiments, performed the experiments, analyzed the data, performed the computation work, prepared figures and/or tables, authored or reviewed drafts of the article, and approved the final draft.
- Reem Alotaibi conceived and designed the experiments, analyzed the data, authored or reviewed drafts of the article, and approved the final draft.
- Haneen Banjar conceived and designed the experiments, authored or reviewed drafts of the article, and approved the final draft.
- Sajjad Karim analyzed the data, authored or reviewed drafts of the article, and approved the final draft.
- Zeenat Mirza performed the computation work, authored or reviewed drafts of the article, and approved the final draft.
- Heba Abusamra analyzed the data, performed the computation work, authored or reviewed drafts of the article, and approved the final draft.
- Manal Al-Attas analyzed the data, authored or reviewed drafts of the article, and approved the final draft.
- Shereen Turkistany analyzed the data, authored or reviewed drafts of the article, and approved the final draft.
- Adel Abuzenadah conceived and designed the experiments, authored or reviewed drafts of the article, and approved the final draft.

### Ethics

The following information was supplied relating to ethical approvals (*i.e.*, approving body and any reference numbers):

The Center of Excellence in Genomics Medical Research bioethical committee approval to carry out the study within its facilities. (License #ot KACST: HA-02-J-003).

### Data Availability

The data and source code are available at Zenodo: Bashanfar, A. (2024). Uncovering Hidden Genetic Risk Factors for Breast and Ovarian Cancers in BRCA-Negative Women: A Machine Learning Approach in the Saudi Population. Zenodo. DOI 10.5281/zenodo.10926611.

### Supplemental Information

Supplemental information for this article can be found online at http://dx.doi.org/10.7717/peerj-cs.1942#supplemental-information.

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
