# Peer review of "Uncovering hidden genetic risk factors for breast and ovarian cancers in BRCA-negative women: a machine learning approach in the Saudi population"

_PeerJ Computer Science, doi:10.7717/peerj-cs.1942_

## Round 0.1 · original submission · Major Revisions

The reviewers find merit in the paper, however, they recommended a major revision before it can be accepted for publication. You are required to address all the comments and suggestions of reviewers and submit a revised manuscript. I will suggest to improve the English language of the manuscript, and correct grammar and typos. Further, One of the reviewers suggested that you cite specific references. You are welcome to add it/them if you believe they are relevant. However, you are not required to include these citations, and if you do not include them, this will not influence my decision.

**Language Note:** The review process has identified that the English language must be improved. PeerJ can provide language editing services - please contact us at [email protected] for pricing (be sure to provide your manuscript number and title). Alternatively, you should make your own arrangements to improve the language quality and provide details in your response letter. – PeerJ Staff

Reviewer 1 ·

Basic reporting

Authors have used Extreme Gradient Boosting (XGBoost) and
Random Forest (RF) algorithms to identify BRCA-related cancer high-risk genes speciûcally
in saudi population.
Work needs certain improvements

Experimental design

Give dataset snopshot in manuscript
How was fine tuning of ML classifier performed.
May abbreviate BRCA at the beginging
was any benchmark dataset used for validation purposes
The algorithnm used for feature extraction needs to be incorporated in the manuscript and discuss the process with the help of any sample sequemce example.
May include the following as well Machine Learning classification algorithms section:PMID: 35215090,PMID: 35552469

The methodology should be reproducable to used.
English needs corrections and improvments at no of places.
Authors must add the contribution statement at the begining added in detailed manner.

The above points must be addressed before finalizing about its publication

Validity of the findings

May add differentiating section at the end of the results which shall describe how this work is different and outperforms the exisitng methods. May mention the bench mark methods as well.

Reviewer 2 ·

Basic reporting

a) Please check and correct for uppercase and lowercase typos in the entire manuscript.
b) It is suggested to improve the figure resolutions ~300DPI.
c) Improve introduction section by adding GLOBOCAN 2022 statistics.

Experimental design

d) Instead of giving a snapshot in figure 4, authors can draft a table for the same.
e) Why was Gene ontology (GO) executed? Has the analysis added any value to this methodology?
f) Why didn't the authors validate their findings from KEGG pathways by generating a new regulatory pathway using some other software/tool available?
g) The authors should add a table encapsulating their keyfindings (genes) and mention about their pivotal functions.
h) How does the machine learning pipeline aid in the bioinformatics pipeline?

Validity of the findings

i) The authors can include a differential expressed gene (DEGs) analysis too for the datasets retrieved from TCGA.

Additional comments

a) Improve introduction section by adding GLOBOCAN 2022 statistics.
b) How does the machine learning pipeline aid in the bioinformatics pipeline?
c) The authors can include a differential expressed gene (DEGs) analysis too for the datasets retrieved from TCGA.

---

## Round 0.2 · accepted · Accept

All the comments and suggestions of the reviewers have been properly addressed. The manuscript may be accepted for publication in its current form.

Reviewer 1 ·

Basic reporting

The manuscript may be accepted now as it is. All the points/comments/suggestions raised have been addressed.

Experimental design

no comment

Validity of the findings

no comment

Reviewer 2 ·

Basic reporting

The authors have improved the changes as per my previous suggestions. No further changes or suggestions from my end.

Experimental design

The authors have improved the changes as per my previous suggestions. No further changes or suggestions from my end.

Validity of the findings

The authors have improved the changes as per my previous suggestions. No further changes or suggestions from my end.

Additional comments

The authors have improved the changes as per my previous suggestions. No further changes or suggestions from my end.